# Outcome in Patients with Partial and Full-Thickness Cheek Defects following Free Flap Reconstruction—A Multicentric Analysis of 47 Cases

**DOI:** 10.3390/jcm9061740

**Published:** 2020-06-04

**Authors:** Stefan Janik, Rachelle Eljazzar, Muhammad Faisal, Stefan Grasl, Erich Vyskocil, Brett A. Miles, Markus Brunner, Rudolf Seemann, Boban M. Erovic

**Affiliations:** 1Department of Otolaryngology, Head and Neck Surgery, Medical University of Vienna, 1090 Vienna, Austria; stefan.janik@meduniwien.ac.at (S.J.); stefan.grasl@meduniwien.ac.at (S.G.); erich.vyskocil@meduniwien.ac.at (E.V.); markus.brunner@meduniwien.ac.at (M.B.); 2Department of Otolaryngology, Head and Neck Surgery, Icahn School of Medicine at Mount Sinai, New York, NY 10029, USA; reljazzar@tulane.edu (R.E.); Brett.Miles@mountsinai.org (B.A.M.); 3Institute of Head and Neck Diseases, Evangelical Hospital Vienna, 1180 Vienna, Austria; maxfas@live.com (M.F.); r.seemann@ekhwien.at (R.S.); 4Department of Head and Neck Surgery, Shaukat Khanum Memorial Cancer Hospital and Research Centre, 54000 Lahore, Pakistan

**Keywords:** through-and-through defect, free flap reconstruction, functional outcome, complications, cheek

## Abstract

The objective of this study was to evaluate whether the extent of tumor resection and free flap reconstruction influences functional outcome and complications in patients with solid malignancies of the cheek. Therefore, we retrospectively assessed recipient site complications and functional outcomes in 47 patients with solid malignancies of the cheek who underwent either partial (*n* = 30; 63.8%) or full-thickness (*n* = 17; 36.2%) cheek resection with free flap reconstruction. Complications occurred in 12 (70.6%) patients after full thickness resections with creation of through-and-through defects compared to 14 (70.6%) patients with partial defects (*p* = 0.138). Among those 26 patients (55.3%), major recipient site complications, like development of salivary fistula or free flap loss, were observed in 10 (21.3%) and 2 (4.3%) cases, respectively, while minor complications, like wound dehiscence and local infections, were found in 14 (29.8%) and 9 (19.1%) patients. Complications were noticed particularly after reconstruction of suborbital defects (69.2%; *p* = 0.268), of which occurrence of salivary fistulae was the most common (46.2%; *p* = 0.035). Similarly, functional outcomes including oral incompetence, ectropion, and trismus were not affected by the extent of resection (*p* = 0.766). However, oral incompetence was higher in patients with tumors originating from the oral cavity (*p* = 0.020) and after the performance of mandibulectomy (*p* = 0.003). Overall, there was no difference in functional outcome or recipient site morbidity between tumor resections resulting in full-thickness and partial defects.

## 1. Introduction

In addition to trauma and burns, resection of malignancy represents the main cause for cheek defects requiring reconstruction [1,2]. Malignancies can arise from the oral mucosa or the skin with infiltration of the submucosa, loose connective tissue, and mimetic muscles of the cheek [3,4]. Occasionally, these malignancies can infiltrate or even arise from neighboring structures, like the nasal cavity or the parotid gland.

Squamous cell carcinomas (SCCs) originating from the oral mucosa or skin are the most common tumor entity of the cheek and surgical tumor resection represents the mainstay of therapy [5,6]. Depending on tumor size and level of invasiveness, extended resections may end up in partial or even though-and-through defects, requiring more complex reconstructions with free flaps [4,7]. Recent publications regarding buccal SCCs further indicate that more extensive tumor resections are even associated with improved oncologic outcome [3,8].

With respect to functional and aesthetic aspects, the cheek region represents the major part of the lateral facial unit that is essential for mimic, facial expression, and maintenance of oral competence [4]. Large composite defects of the cheek also often result in deficits of the facial nerve, further adding to the aesthetic and functional morbidity of the resection. Hence, cheek reconstruction is challenging and requires thorough knowledge of anatomic subsites and danger zones, corresponding function, and available reconstructive techniques to achieve functionally and aesthetically satisfactory results [4].

Although studies have already reported on improved oncological outcome after more extensive resections of tumors of the cheek, functional endpoints, particularly after free flap reconstruction and associated recipient site morbidity, have not been addressed so far. It was therefore the purpose of this retrospective, multicentric study to evaluate the effect of cheek resection and free flap reconstruction on the functional outcomes (I) and associated complications (II).

## 2. Material and Methods

### 2.1. Study Cohort

We conducted a retrospective, multicenter chart review of patients with solid malignancies originating in the cheek who underwent tumor resection and free flap reconstruction between 2012 and 2017. Patients were treated at the Department of Otorhinolaryngology, Head and Neck Surgery, Medical University of Vienna, Austria (Center 1), the Institute of Head and Neck Diseases, Evangelical Hospital Vienna (Center 2), the Department of Otolaryngology, Head and Neck Surgery, Icahn School of Medicine at Mount Sinai, New York, USA (Center 3), and the Department of Head and Neck Surgery, Shaukat Khanum Memorial Cancer Hospital and Research Centre, Lahore, Pakistan (Center 4). All surgeries were performed by experienced head and neck surgeons who had completed previously head and neck fellowships.

Data of potential patients were provided by attending centers that were further evaluated regarding appropriateness by two authors individually (S.J., B.E.). Patients had to fulfill the following inclusion criteria: carcinomas with infiltration of the cheek, including oral carcinomas, carcinomas of the nasal cavity, skin and parotid gland (I), solid carcinomas (II), radical tumor resection with free flap reconstruction (III), and primary or recurrent carcinomas (IV). Finally, 47 patients were eligible for inclusion and analysis. Ethical approval was obtained from the Ethics Committee of the participating centers prior to enrolment.

### 2.2. Classification of Cheek Carcinomas

From the aesthetic point of view, the cheek region can be divided into three overlapping aesthetic zones including the suborbital (I), the preauricular (II), and the bucco-mandibular (III) zone, as illustrated in Figure 1 [9]. Anatomically, the cheek consists (from inside to outside) of buccal mucosa, submucosa, loose connective tissue, muscles, the parotid gland in zone II and III, and the skin. Depending on the extent of resection, and consequently, the depth of defect, patients were dichotomized into patients with partial or through-and-through (full) defects.

### 2.3. Free Flap Reconstruction

Depending on the size and depth of the defect, cutaneous, myocutaneous or osteocutaneous flaps were used. In cases with partial defects, harvested skin paddles were used for the inner lining of the oral cavity or for reconstruction of the skin if required. Conversely, in patients with through-and-through defects, split or full thickness skin grafts were used for the inner lining, while harvested skin paddles were used for the outer lining (Figure 2).

### 2.4. Complications and Functional Outcome

We assessed recipient site complications and functional outcomes as main endpoints of the study within the first 6 to 12 months after surgery and free flap reconstruction. Recipient site complications were further classified either as minor complications including wound dehiscence and local infection or major complications including salivary fistula and free flap failure. Presence of ectropion, oral incompetence and trismus were used as functional endpoints. All outcomes were rated by treating head and neck surgeons.

### 2.5. Statistical Methods

Statistical analyses were performed using SPSS software (version 22; IBM SPSS Inc., Chicago, IL, USA). Unless otherwise specified, data in the results section are shown as median ± standard deviation. Chi-square test was used to assess associations between nominal variables. In cases with expected cell counts below 5, *p*-values of Fisher’s exact value were reported. Moreover, the unpaired student’s *t*-test was used to compare means of normally distributed variables of two independent groups. 

## 3. Results

### 3.1. Study Cohort

Forty-seven patients, including 29 men (61.7%) and 18 women (38.3%), with a median age of 64 ± 15.2 years (range: 30y–93y), were included in this retrospective, multicentric analysis, all of whom underwent tumor resection and free flap reconstruction of the cheek. SCC was the predominant tumor histology (*n* = 38; 80.9%) followed by adenoid cystic carcinoma (ACC; *n* = 3; 6.4%), sarcoma (*n* = 2; 4.3%), melanoma (*n* = 2; 4.3%), merkel cell carcinoma (MCC; *n* = 1; 2.1%), and malignant adnexal skin tumor (*n* = 1; 2.1%). Malignancies originated from buccal mucosa, oral cavity, parotid gland, nasal cavity and skin in 22 (46.8%), 10 (21.3%), 4 (8.5%), 3 (6.4%) and 2 (4.3%) cases, respectively. With respect to aesthetic zones, tumors were located predominantly at zone I, II, and III in 13 (27.7%), 11 (23.4%), and 23 (48.9%) cases, respectively (Table 1).

### 3.2. Tumor Characteristics

Patients had three T1 (6.4%), eleven T2 (23.4%), twelve T3 (25.5%), and twenty-one T4 (44.7%) tumors, respectively, with a median tumor size of 3.9 ± 1.9 cm (range: 1.0–8.8 cm). At initial presentation, 30 (63.8%), 6 (12.8%), and 7 (14.9%) patients had N0, N1 and N2 disease, while cervical lymph node classification (Nx) was unknown in 4 patients (8.5%). Altogether, we had 3 stage I (6.4%), 8 stage II (17.0%), 11 stage III (23.4%), and 25 stage IV (53.2%) malignancies (Table 1).

### 3.3. Surgical Resection

Radical tumor resection created partial and through-and-through defects in 30 (63.8%) and 17 (36.2%) patients, respectively. Socio-demographic characteristics, including male to female ratio (21:9 vs. 8:9), age (65.1 ± 15.8y vs. 60.7 ± 14.0y), and body-mass-index (25.1 ± 4.4 kg/m^2^ vs. 24.5 ± 4.6 kg/m^2^) did not significantly differ in patients with partial and through-and-through defects, respectively (*p* = 0.211; *p* = 0.388; *p* = 0.677). Moreover, T-classification (*p* = 0.901), N-classification (*p* = 0.372), and AJCC tumor stage (*p* = 0.492), did also not significantly differ between both groups (Table 1).

ND was performed in 80.9% of patients. Level I–III, I–IV, II–IV, and II–III ND were done in 13 (27.7%), 9 (19.1%), 8 (17.0%), and 4 (8.5%) cases, respectively. In two patients (4.2%), the extent of ND was not indicated. Maxillectomy was necessary to perform in 19 out of 47 patients (40.4%), including partial-, hemi-, and total maxillectomy in 5 (10.6%), 9 (19.1%), and 5 (10.6%) cases, respectively. Otherwise, partial and total mandibulectomy was done in 11 (23.4%) and 1 patient (2.1%). Primary tumor resection was further accompanied by partial glossectomy and orbital exenteration in 4 (8.5%) and 3 patients (6.4%).

### 3.4. Free Flap Reconstruction

The radial forearm free flap (RFFF) was most commonly used for cheek reconstruction (*n* = 15; 31.9%) followed by anterolateral thigh (ALT) flap (*n* = 13; 27.7%), scapular/parascapular free flap (*n* = 10; 21.3%), FFF (fibula free flap; *n* = 6; 12.8%), supraclavicular free flap (*n* = 2; 4.3%), and serratus anterior free flap (SAFF; *n* = 1; 2.1%), respectively. Altogether, cutaneous, myocutcaneous and osteocuteanous free flaps were harvested in 19 (40.4%), 12 (25.5%) and 16 (34.0%) cases, respectively (Table 2). As indicated in Table 2, RFFF was most commonly used for reconstruction of one-layer skin or mucosal defects (14 out of 19; 73.7%), while the ALT flap was mostly used as myocutaneous flap (10 out of 12; 83.3%), and the scapular/parascapular free flap for bone reconstruction (10 out of 16; 62.5%) (Table 2). In eight patients (17.0%), free flaps were oversized and too bulky. Bulkiness of the free flap occurred particularly in zone I defects (30.8% vs. 11.8%), more likely in through-and-through defects (29.4% vs. 10.0%), and after harvest of free scapular/parascapular free flaps (40.0% vs. 10.8%). However, differences failed to reach statistical significance (*p* = 0.288; *p* = 0.118; *p* = 0.331) and revision surgery with thinning of the free flap was performed in six out of eight patients in order to optimize final cosmetic results.

We had two losses of free flaps resulting in a free flap success rate of 95.7%, while flap revision due to venous congestion was necessary in three (6.4%) cases. In those two cases with flap loss, an ALT and a latissimus dorsi flap were used for revision surgery.

Donor site complications, which have been of minor concern, were observed in six patients (12.8%) who experienced wound dehiscence.

### 3.5. Complications

Recipient site complications occurred in 26 (55.3%) patients, which was not statistically significant different between patients with partial compared to those with through-and-through defects (46.7% vs. 70.6%; *p* = 0.138). Wound dehiscence, formation of salivary fistula, local infections and free flap failure occurred in 14 (29.8%), 10 (21.3%), 9 (19.1%), and 2 (4.3%) cases, respectively, but did not significantly differ between both groups (Table 3).

However, with regards to affected aesthetic zones, we observed formation of salivary fistula particularly in 46.2% of zone I defects, which was significantly higher compared to 9.1% and 13.0% in zone II and III defects, respectively (*p* = 0.035). It is noteworthy to mention that extent of resection (*p* = 0.136), performance of maxillectomy (*p* = 0.496) or mandibulectomy (*p* = 1.000), T-classification (*p* = 0.751) or size of defect (*p* = 0.145) had no significant impact on the development of salivary fistulae.

### 3.6. Functional Outcome

Oral incompetence, ectropion, and trismus occurred in nine (19.1%), eight (17.0%), and six patients (12.8%), respectively. Again, the extent of resection had no significant impact on the development of any functional impairment (Table 3). Nonetheless, solely patients with malignancies originating of the buccal mucosa and oral cavity suffered from oral incompetence (22.7% and 66.7%; *p* = 0.020). Performance of mandibulectomy (*p* = 0.003), but not maxillectomy (*p* = 0.064), affected significantly oral competence, which was otherwise significantly associated with the occurrence of trismus (*p* = 0.009). In addition, the size of used free flaps was 14.7 ± 5.1 cm in patients with oral incompetence, which was significantly larger compared to 7.4 ± 2.7 cm in patients with oral competence (*p* = 0.008), while primary tumor size did not significantly differ (4.4 ± 1.4 vs. 3.5 ± 1.9 cm; *p* = 0.206) (Table 3).

## 4. Discussion

We have analyzed clinical outcome of 47 patients with solid malignancies of the cheek that underwent radical tumor resection with creation of partial or through-and-through defects and free flap reconstruction. Within our study, SCCs were the predominant histologic subtype (80.9%) and malignancies mainly originated from the oral cavity (80.8%). This is consistent with the literature, reporting mostly on oral carcinomas and rarely on skin carcinomas, requiring cheek reconstruction with free flaps following oncological resections [3,4]. Nonetheless, our data further display the great diversity of tumors affecting the cheek region that may hamper analysis of more homogenous subgroups with large patient numbers.

In solid malignancies, and for patients with SCCs in particular, surgical tumor resection with adjuvant therapy in selected cases represents the most frequent treatment modality [5,6,10]. However, despite radical surgical resection, recurrence rates range from 45.0% to 80.0% in patients with buccal SCCs [11,12]. Several authors assume that the absence of “real” anatomic boundaries limiting tumor growth and spread might contribute to the high rate of recurrences [11,13]. This prompted Ren ZH and coworkers (2017) to perform a more extensive resection of functional anatomic buccal units to achieve a better oncologic outcome [3]. In fact, they analyzed data of 127 patients with buccal SCCs reporting on significantly better 2-year overall survival (OS: 83.3% vs. 60.1%) and disease-free survival (disease-free survival (DFS): 76.6% vs. 51.9%) in patients undergoing more extensive unit resection compared to conventional surgery [3].

Although oncologic principles must supersede reconstructive desires [7], we were particularly interested in knowing how the extent of resection impacts functional outcome. Oral incompetence represented the main functional complication occurring in nine (19.1%) patients followed by occurrence of ectropion and trismus in eight (17.0%), and six patients (12.8%), respectively. This is in line with the results of other publications, reporting on problems with oral incompetence in 4.8% up to 40% of patients with cheek carcinomas [8,14,15]. It is noteworthy to mention that the extent of resection had no significant impact on functional outcome in our cohort, which is in accordance to the work of Ren ZH et al. (2017). The authors assume that insignificantly changed functional outcomes in patients with conventional surgery compared to more extensive unit resections, have resulted from the loss of function of preserved structures secondary to induction of fibrosis and loss of functional adjacent structures/attachments by tumor resection [3]. However, we found a strong association between oral incompetence and trismus in patients after mandibulectomy. This indicates that functional outcomes more likely depend on the preservation of certain anatomic structures and chosen surgical approach than on the depth of defect.

Recipient site complication rate was 55.3% (*n* = 26), of which wound dehiscence was the most common complication occurring in 29.8% (*n* = 14) of cases. Although the majority of complications were of minor concern, 10 patients developed salivary fistulae (21.3%) that occurred significantly more often in suborbital zone I defects (*p* = 0.035). This is in accordance to former studies reporting on fistula rates of 4.3% to 27.3% of patients [8,14,16,17,18]. The development of fistulae is characteristic for maxillary reconstruction and occurs typically near to the medial canthus (zone I) due to breakdown of suture lines [16]. In alignment to that, we observed the highest rate of wound dehiscence (38.5%) in patients with zone I defects compared to 18.2% and 30.4% in zone II and III defects, respectively. We assume that gravity causes drag down of cheek skin by bulky flaps, resulting in a significant scarring, wound break, and occasionally creation of salivary fistula and ectropion with epiphora. Depending on how much the wound break and ectropion have progressed either a lateral tarsorrhaphy with a bone anchored suture, particularly at the orbital rim, or additionally to the tarsorrhaphy, a split or full thickness skin graft should be placed below the lower lid to gain more skin volume. By doing so, additional skin may prevent wound break and subsequently creation of salivary fistula as well as development of ectropion and epiphora. We think this is the best option to prevent significant aesthetic and functional sequelae in suborbital zone I defects.

Until now, a number of different free flaps have been described for cheek reconstruction including the ALT [19], RFFF [19], FFF [20] and the scapular/parascapular free flap [21]. Recently, the versatility of the SAFF has been demonstrated for general head and neck reconstruction, and in particular, for cheek and tongue reconstruction [22,23]. In our study, the RFFF was used in 31.9% (n = 15) of cases followed by the ALT free flap and the scapular/parascapular free flap in 27.7% (*n* = 13) and 21.3% (*n* = 10) of patients, respectively. Among those, the RFFF was mainly harvested for cutaneous reconstruction (73.7%), the ALT flap for reconstruction of myocutaneous defects (83.3%), and the scapular/parascapular free flap for bone reconstruction (62.5%). Our flap survival rate was 95.7%, which is comparable to 95% to 96% reported in former studies [17,19].

We believe that the strength of this study lies in the analysis of functional outcomes as well as complications. We see three limiting factors: first, the retrospective study design bears an inherent risk of information and selection bias. Second, the heterogeneity of our cohort with solid malignancies originating from different parts of the cheek allows only limited conclusions. Third, the lack of standard measures for functional outcomes in head and neck oncology [24] and the fact that functional and aesthetic outcomes have been rated by treating head and neck surgeons as opposed to patient reported outcomes, represent further limitations.

## 5. Conclusions

More extensive resections with creation of through-and-through defects did not automatically correlate with worse functional outcome or higher recipient site complications. Nonetheless, especially suborbital zone I defects carried an increased risk of wound dehiscence and formation of salivary fistula, and requires, therefore, particular consideration and meticulous reconstruction. However, further prospective studies with homogenous patient cohorts are necessary to define and identify additional factors that may contribute to the functional outcome of patients with carcinomas of the cheek.

## Figures and Tables

**Figure 1 jcm-09-01740-f001:**
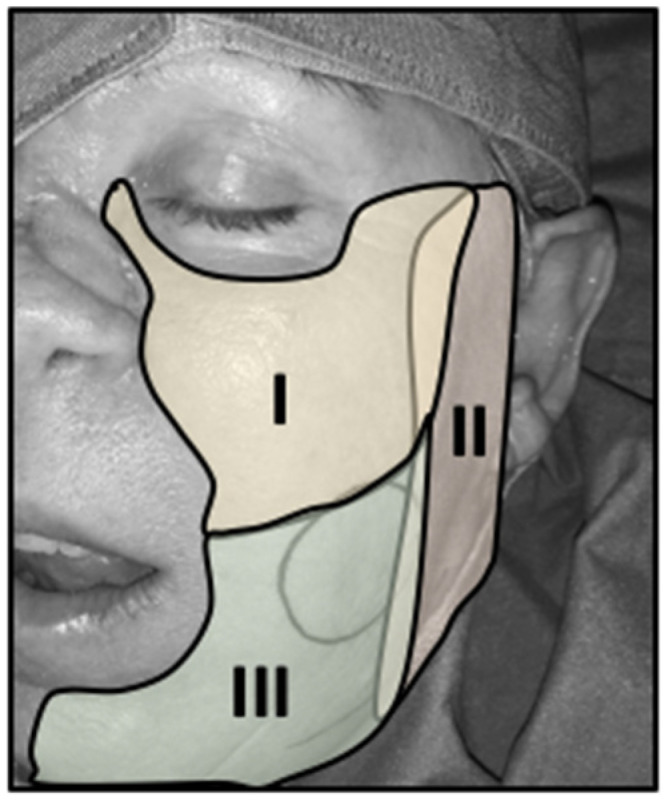
Aesthetic zones of the cheek. The cheek can be divided into three partially, overlapping aesthetic zones, comprising the suborbital (**I**), the preauricular (**II**), and the bucco-mandibular (**III**) zone [9]. The circle already marks the margins for planed tumor resection of a squamous cell carcinoma that is mainly located in zone III.

**Figure 2 jcm-09-01740-f002:**
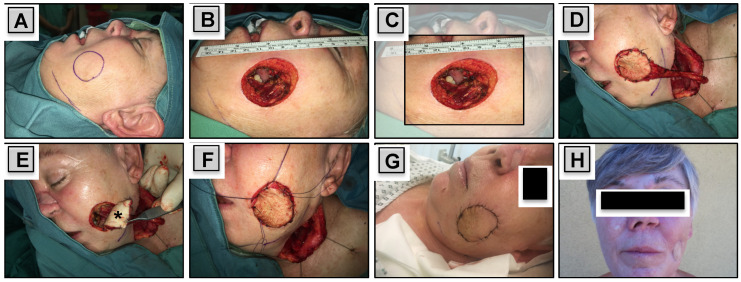
Reconstruction of a Through-and-Through Defect. A 60-year-old female patient experienced regional failure of a sinonasal carcinoma. Radical tumor resection was performed, resulting in creation of a through-and-through defect with 4.0 × 5.0 cm in size (**A**–**C**). A free radial forearm free flap (RFFF) was used for reconstruction. The harvested skin paddle was used for the outer lining (**D**,**F**), while full thickness skin graft of the neck (asterisk) was used for inner lining (**E**). Postoperative (**G**) and 2 year follow up results are shown (**H**). The reconstructed region on the left side of the nasal dorsum (**D**–**G**) represents the former tumor infiltrated area that has been resected and reconstructed.

**Table 1 jcm-09-01740-t001:** Patient Characteristics.

		Depth of Defect		
	Total	Partial	Full	
Variables	*n* (%)	*n* (%)	*n* (%)	*p* ^a^
**Sex**				
Female	18 (38.3)	9 (30.0)	9 (52.9)	
Male	29 (61.7)	21 (70.0)	8 (47.1)	0.211
**Tumor Site**				
Buccal Mucosa	22 (46.8)	12 (40.0)	10 (58.8)	
Oral Cavity	16 (34.0)	12 (40.0)	4 (23.5)	
Parotid Gland	4 (8.5)	4 (13.3)	0 (0)	
Nasal Cavity	3 (6.4)	1 (3.3)	2 (11.8)	
Skin	2 (4.3)	1 (3.3)	1 (5.9)	0.255
**Histology**				
SCC	38 (80.9)	24 (80.0)	14 (82.4)	
ACC	3 (6.4)	3 (10.0)	0 (0)	
Sarcoma	2 (4.3)	1 (3.3)	1 (5.9)	
Melanoma	2 (4.3)	1 (3.3)	1 (5.9)	
MCC	1 (2.1)	1 (3.3)	0 (0)	
Malignant Adnexal Tumor	1 (2.1)	0 (0)	1 (5.9)	0.497
**T—Classification**				
T1	3 (6.4)	2 (6.7)	1 (5.9)	
T2	11 (23.4)	8 (26.7)	3 (17.6)	
T3	12 (25.5)	7 (23.3)	5 (29.4)	
T4	21 (44.7)	13 (43.3)	8 (47.1)	0.901
**N—Classification**				
Nx	4 (8.5)	4 (13.3)	0 (0)	
N0	30 (63.8)	17 (56.7)	13 (76.5)	
N1	6 (12.8)	4 (13.3)	2 (11.8)	
N2	7 (14.9)	5 (16.7)	2 (11.8)	
N3	0 (0)	0 (0)	0 (0)	0.372
**AJCC Tumor Stage**				
Stage I	3 (6.4)	2 (6.7)	1 (5.9)	
Stage II	8 (17.0)	6 (20.0)	2 (11.8)	
Stage III	11 (23.4)	7 (23.3)	4 (23.5)	
Stage IV	25 (53.2)	15 (50.0)	10 (58.8)	0.896
**Aesthetic Zone**				
Zone I	13 (27.7)	7 (23.3)	6 (35.3)	
Zone II	11 (23.4)	9 (30.0)	2 (11.8)	
Zone III	23 (48.9)	14 (46.7)	9 (52.9)	0.334

ACC, adenoid cystic carcinoma; AJCC, American Joint Committee on Cancer; MCC, merkel cell carcinoma; n, number of patients; N—Classification, cervical lymph node classification; Nx, unknown cervical lymph node status; SCC, squamous cell carcinoma; T—Classification, tumor size classification; *p*, *p*-value; ^a^ chi-square test.

**Table 2 jcm-09-01740-t002:** Free flap Reconstruction.

		Used Free Flaps					
Variables	Total	RFFF	ALT	Scapular/ Parascapular	FFF	Supraclav. Free Flap	SAFF
**Depth of Defect**							
Partial	30 (63.8)	8 (26.7)	11 (36.7)	5 (16.7)	4 (13.3)	2 (6.7)	0 (0)
Full	17 (36.2)	7 (41.2)	2 (11.8)	5 (29.4)	2 (11.8)	0 (0)	1 (5.9)
**Type of Reconstruction**							
Cutaneous	19 (40.4)	14 (73.7)	3 (15.8)	0 (0)	0 (0)	2 (10.5)	0 (0)
Myocutaneous	12 (25.5)	1 (8.3)	10 (83.3)	0 (0)	0 (0)	0 (0)	1 (8.3)
Osteocutaneous	16 (34.0)	0 (0)	0 (0)	10 (62.5)	6 (37.5)	0 (0)	0 (0)
**Aesthetic Zone**							
Zone I	13 (27.7)	4 (30.4)	4 (30.4)	4 (30.4)	0 (0)	0 (0)	1 (7.7)
Zone II	11 (23.4)	1 (9.1)	4 (36.4)	3 (27.3)	3 (27.3)	0 (0)	0 (0)
Zone III	23 (48.9)	10 (43.5)	5 (21.7)	3 (13.0)	3 (13.0)	2 (8.7)	0 (0)
**Total**	47 (100)	15 (31.9)	13 (27.7)	10 (21.3)	6 (12.8)	2 (4.3)	1 (2.1)

ALT, anterolateral thigh flap; FFF, fibula free flap; RFFF, radial forearm free flap; SAFF, serratus anterior free flap.

**Table 3 jcm-09-01740-t003:** Complications and functional outcome according to depth of defect and aesthetic zone.

	Depth of Defect				Aesthetic Zones			
	Total	Partial	Full		Zone I	Zone II	Zone III	
Complications	*n* (%)	*n* (%)	*n* (%)	*p* ^a^	*n* (%)	*n* (%)	*n* (%)	*p* ^a^
**Recipient Site**								
No	21 (44.7)	16 (53.3)	5 (29.4)		4 (30.8)	7 (63.6)	10 (43.5)	
Yes	26 (55.3)	14 (46.7)	12 (70.6)	0.138	9 (69.2)	4 (36.4)	13 (56.5)	0.268
Infection	9 (19.1)	5 (16.7)	4 (23.5)	0.704	3 (23.1)	0 (0)	6 (26.1)	0.178
Dehiscence	14 (29.8)	6 (20.0)	8 (47.1)	0.095	5 (38.5)	2 (18.2)	7 (30.4)	0.554
Fistula	10 (21.3)	4 (13.3)	6 (35.3)	0.136	6 (46.2)	1 (9.1)	3 (13.0)	**0.035**
Flap Failure	2 (4.3)	1 (3.3)	1 (5.9)	1	1 (7.7)	1 (9.1)	0 (0)	0.362
Functional Outcome								
**Functional Impairment**								
No	29 (61.7)	19 (63.3)	10 (58.8)		9 (69.2)	7 (63.6)	13 (56.5)	
Yes	18 (38.3)	13 (36.7)	7 (41.2)	0.766	4 (30.8)	4 (36.4)	10 (43.5)	0.744
Ectropion	8 (17.0)	5 (16.7)	3 (17.6)	1	2 (15.4)	2 (18.2)	4 (17.4)	0.981
Oral Incompetence	9 (19.1)	4 (13.3)	5 (29.4)	0.252	2 (15.4)	1 (9.1)	6 (26.1)	0.46
Trismus	6 (12.8)	3 (10.0)	3 (17.6)	0.653	2 (15.4)	1 (9.1)	3 (13.0)	0.898

Full, Through-and-Through defects; *n*, number of patients; *p*, *p*-value; ^a^ chi-square test.

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
