# Peer review of "Outcome in Patients with Partial and Full-Thickness Cheek Defects following Free Flap Reconstruction—A Multicentric Analysis of 47 Cases"

_jcm, 2020, doi:10.3390/jcm9061740_

Round 1
Reviewer 1 Report
This manuscript describes about the analysis of functional outcome and complications of tumor resection and free flap reconstruction in patients with solid malignancies of the cheek. As the authors say in the report, the cohort consists of solid malignancies originating from different parts of the cheek, which makes it difficult to draw substantial conclusions. However, the main research findings of this paper that the extent of resection had no significant impact on functional outcome will be important for clinical practice of buccal malignancies, because though-and-through resection of the cheek and reconstruction would not be common case for many head and neck surgeons.
P.2 L.78-79: “From the aesthetic point of view, the cheek region can be divided into three overlapping aesthetic zones….” Head and neck surgeons may not be familiar with this classification and it is helpful to illustrate the three zones in a simple figure.
P7.L227-229: “The development of fistulae is characteristic for maxillary reconstruction and occurs typically near to the medial canthus (zone I) due to break down of suture lines.” Is there any method to overcome the vulnerability of this part?
Author Response
Reply to the Review Report (Reviewer 1)
This manuscript describes about the analysis of functional outcome and complications of tumor resection and free flap reconstruction in patients with solid malignancies of the cheek. As the authors say in the report, the cohort consists of solid malignancies originating from different parts of the cheek, which makes it difficult to draw substantial conclusions. However, the main research findings of this paper that the extent of resection had no significant impact on functional outcome will be important for clinical practice of buccal malignancies, because though-and-through resection of the cheek and reconstruction would not be common case for many head and neck surgeons.
Comment 1: P.2 L.78-79: “From the aesthetic point of view, the cheek region can be divided into three overlapping aesthetic zones….” Head and neck surgeons may not be familiar with this classification and it is helpful to illustrate the three zones in a simple figure.
ANSWER and CHANGES 1: As requested, we created an additional figure illustrating the three different and partially overlapping aesthetic zones.
- Text was adapted as follows – Page 2, Lines 86-88: “From the aesthetic point of view, the cheek region can be divided into three overlapping aesthetic zones including the suborbital (I), the preauricular (II), and the bucco-mandibular (III) zone as illustrated in Figure 1.9 “
- Figure 1 and corresponding figure legend (Page 3) were added:
Figure 1. Aesthetic zones of the cheek. The cheek can be divided into three partially, overlapping aesthetic zones, comprising the suborbital (I), the preauricular (II), and the bucco-mandibular (III) zone.9 The circle already marks the margins for resection of a squamous cell carcinoma, which is mainly located in zone III.
Comment 2: P7.L227-229: “The development of fistulae is characteristic for maxillary reconstruction and occurs typically near to the medial canthus (zone I) due to break down of suture lines.” Is there any method to overcome the vulnerability of this part?
ANSWER 2: Thank you very much for this important remark: in fact, this is the most vulnerable part postoperatively and during the rehabilitation phase! Due to gravity the flap drags the cheek skin down resulting in a significant scarring, wound break, fistula and ectropion with epiphora. Depending how much the wound break and ectropion have progressed either a lateral tarsorrhaphy with a bone anchored suture, particularly at the orbital rim, or additional to the tarsorrhaphy, a split or full thickness skin graft should be placed below the lower lid to gain more skin volume. By doing so, additional skin prevents wound break and development of ectropion and epiphora. We think this is the best option to prevent significant aesthetic and functional sequelae.
CHANGES 2: Discussion was adapted as follows: Discussion (Page 8,9; Lines 251 – 259): “We assume that gravity causes drag down of cheek skin by bulky flaps resulting in a significant scarring, wound break, and occasionally creation of salivary fistula and ectropion with epiphora. Depending how much the wound break and ectropion have progressed either a lateral tarsorrhaphy with a bone anchored suture, particularly at the orbital rim, or additional to the tarsorrhaphy, a split or full thickness skin graft should be placed below the lower lid to gain more skin volume. By doing so, additional skin may prevent wound break and subsequently creation of salivary fistula as well as development of ectropion and epiphora. We think this is the best option to prevent significant aesthetic and functional sequelae in suborbital zone I defects. “

Reviewer 2 Report
Janik et al. present a multicentric study which included 47 patients out of 3 countries to compare the functional outcome of patients after partial and full-thickness cheek defects with a free flap reconstruction.
It is good conducted study and it is well written.
Some concerns are listed below:
- How many different surgeons operated the patients? Who did surgery: Chief of the department/consultant/ or?
- It remains unclear at which time point after surgery you evaluated the functional outcome. In my opinion it could be improve the manuscript if you add a table with follow up of progress (for example after 6 weeks, 6 month, 12 month, 24 month)
- Did the patients received any kind of physiotherapy or speech therapy (including swallowing training)? Was there a different between the two groups?
- For a further study idea: In my opinion it could be useful to give a questionnaire to patients for asking how they are satisfied with swallowing, etc. at different time points after surgery. This would complete the aim of your study.
Author Response
Reply to the Review Report (Reviewer 2)
Janik et al. present a multicentric study which included 47 patients out of 3 countries to compare the functional outcome of patients after partial and full-thickness cheek defects with a free flap reconstruction. It is good conducted study and it is well written.
Some concerns are listed below:
Comment 3: How many different surgeons operated the patients? Who did surgery: Chief of the department/consultant/ or?
ANSWER 3: We are grateful for this valuable comment. Surgeries were performed or supervised by three head and neck fellowship-trained surgeons.
CHANGES 3: Material and Methods were adapted: Page 2, Lines 75-76: “All surgeries were performed by experienced head and neck surgeons who had completed previously head and neck fellowships.“
Comment 4: It remains unclear at which time point after surgery you evaluated the functional outcome. In my opinion it could improve the manuscript if you add a table with follow up of progress (for example after 6 weeks, 6 month, 12 month, 24 month)?
ANSWER 4: This is an absolutely relevant and important issue. Indeed, functional outcomes were evaluated within the first 6 - 12 months after resection and free flap reconstruction.
CHANGES 4: Text within Material and Methods was corrected: Page 3, Line 98-99: “We assessed recipient site complications and functional outcomes as main endpoints of the study within the first 6 to 12 months after surgery and free flap reconstruction. “
Comment 5: Did the patients receive any kind of physiotherapy or speech therapy (including swallowing training)? Was there a difference between the two groups?
ANSWER 5: Patients did not receive any specific speech therapy automatically. Additional therapies, like speech therapy or physiotherapy, were performed solely in patients with complaints. Therefore, we do not know whether automatic speech therapy may have an impact on functional outcome.
CHANGES 5: None
Comment 6: For a further study idea: In my opinion it could be useful to give a questionnaire to patients for asking how they are satisfied with swallowing, etc. at different time points after surgery. This would complete the aim of your study.
ANSWER and CHANGES 6: We totally agree with Reviewer 2 that self-assessment of patients’ well-being and especially of swallowing, speech and chewing are of certain interest. Indeed, we are already performing a follow-up study including head and neck cancer patients undergoing free flap reconstruction whom will be asked to accomplish the 4th version of the University of Washington Quality of Life Questionnaire (UW-QOL).

Reviewer 3 Report
I want to thank the authors for their work on outcomes following cheek reconstruction. The author's work is valuable and overall well composed. Their study will add outcome data that will contribute to the currently available literature on that topic. It is a well written manuscript.
You may want to consider the following points:
Abstract:
l23: Well, given the low number of cases, it may be better to provide absolute numbers of patients who had complications
Introduction is fine
Materials and Methods:
In case of any disagreement, cases were reevaluated until consensus could be achieved ? I would say that there should not be any disagreement about a case matching the inclusion criteria or not. In that case, your inclusion criteria may be too broad. Please remove this sentence.
Figure 1. Please clarify: Patient experienced regional failure of a sinonasal carcinoma. Given the images and text alone this does make little sense. Provide a little more detail here. Additionally, panel H shows front view of the patient. All of the other panels show a lateral view. Why is that?
Results
3.2. tumor characteristics
please write numbers (e.g. three instead of 3) before T classification to make it less confusing
3.4. Complications
What about donor site complications? This may be of interest as well.
Line 171: do you report on both major and minor complications? Make this more clear, and please also add this info to the abstract
Discussion
207: Overall survival (OS), Disease free survival (DFS). The same for SCC line 194. Provide the full term once. Check throughout the manuscript
Line 240: I disagree that multi centric studies are particularly great for studies such as yours. One of the major influencer variables of surgical (especially in surgical oncology) outcomes is the surgeon, hospital, infrastructure. all of the potentially confounding variables cannot be taken into consideration in your study (which is fine per se). But i would not state that this is a strength of your study.
Conclusion
highly-required? replace with "necessary"
Your conclusion is not great. Please identify some of the findings in your manuscript that you think are really of interest. You can always conclude that there is more to be explored and studied.
Author Response
Reply to the Review Report (Reviewer 3)
I want to thank the authors for their work on outcomes following cheek reconstruction. The author's work is valuable and overall well composed. Their study will add outcome data that will contribute to the currently available literature on that topic. It is a well written manuscript. You may want to consider the following points:
Comment 7: Abstract: l23: Well, given the low number of cases, it may be better to provide absolute numbers of patients who had complications
ANSWER and CHANGES 7: As recommended by Reviewer 3, Abstract was adapted as follows:
- Abstract: Page 1, Lines 17-33: “ (…) Results: Complications occurred in 12 (70.6%) patients after full thickness resections with creation of through-and-through defects compared to 14 (70.6%) patients with partial defects (p=0.138). Full thickness resections with creation of through-and-through defects were not associated with significantly higher complication rates (70.6% vs. 46.7%; p=0.138) compared to partial defects. Recipient site complications occurred in 26 3%of patients (55.3%) and were noticed most likely after reconstruction of suborbital defects (69.2%; p=0.268) of which occurrence of salivary fistulae was the most common (46.2%; p=0.035). (…)“
Comment 8: Materials and Methods: In case of any disagreement, cases were reevaluated until consensus could be achieved? I would say that there should not be any disagreement about a case matching the inclusion criteria or not. In that case, your inclusion criteria may be too broad. Please remove this sentence.
ANSWER and CHANGES 8: We totally agree with the reviewer and the sentence was removed as suggested.
- Page 2, Lines 81-82: “In case of any disagreement, cases were reevaluated until consensus could be achieved. “
Comment 9: Figure 1. Please clarify: Patient experienced regional failure of a sinonasal carcinoma. Given the images and text alone this does make little sense. Provide a little more detail here. Additionally, panel H shows front view of the patient. All of the other panels show a lateral view. Why is that?
ANSWER 9: As shown in Figure 2E (former 1E), the reconstructed region on the left side of the nasal dorsum represents the former tumor infiltrated area that has been resected and reconstructed. Therefore, we classified the SCC localized in the subcutaneous tissue of the left cheek as regional failure.
Regarding the last raised point of Reviewer 3, we are showing in panel H a frontal view of the patient intending to provide a better impression of the overall symmetry and subsequently aesthetic outcome of the patient.
CHANGES 9: Figure legend of Figure 2 was completed: “The reconstructed region on the left side of the nasal dorsum (D,E,G) represents the former tumor infiltrated area that has been resected and reconstructed. “
Comment 10: Results: 3.2. tumor characteristics. please write numbers (e.g. three instead of 3) before T classification to make it less confusing.
ANSWER and CHANGES 10: The corresponding paragraph was changed as follows:
- Page 6, Lines 144-145: “Patients had 3 three T1 (6.4%), 11 eleven T2 (23.4%), 12 twelve T3 (25.5%), and 21 twenty-one T4 (44.7%) tumors, respectively, with a median tumor size of 3.9 ± 1.9 cm (range: 1.0 - 8.8 cm).”
Comment 11: 3.4. Complications. What about donor site complications? This may be of interest as well.
ANSWER and CHANGES 11: Once more we totally agree with the reviewer. We also assessed donor site complications that have been of minor concern (wound dehiscence) and occurred in only 6 patients (12.8%). Following sentence was adapted in the Result section: Page 6, Lines 182-183: “Donor site complications, which have been of minor concern, were observed in 6 patients (12.8%) who experienced wound dehiscence.“
Comment 12: Line 171: do you report on both major and minor complications? Make this more clear, and please also add this info to the abstract
ANSWER 12: We presented the overall recipient site complication rate, including minor and major complications, if not otherwise specified. Due to the fact that some patients developed both minor and major complications (e.g. wound dehiscence and formation of salivary fistula), we preferred to indicate complication rate for each complication separately within the text. Abstract was modified as requested.
CHANGES 12:
- Abstract: Page 1: “ Results: Complications occurred in 12 (70.6%) patients after full thickness resections with creation of through-and-through defects compared to 14 (70.6%) patients with partial defects (p=0.138). Full thickness resections with creation of through-and-through defects were not associated with significantly higher complication rates (70.6% vs. 46.7%; p=0.138) compared to partial defects. Recipient site complications occurred in 55.3% of patients and were noticed most likely after reconstruction of suborbital defects (69.2%; p=0.268) of which occurrence of salivary fistulae was the most common (46.2%; p=0.035). Among those 26 patients (55.3%), major recipient site complications like development of salivary fistula or free flap loss were observed in 10 (21.3%) and 2 (4.3%) cases, respectively, while minor complications like wound dehiscence and local infections were found in 14 (29.8%) and 9 (19.1%) patients. Complications were noticed particularly after reconstruction of suborbital defects (69.2%; p=0.268) of which occurrence of salivary fistulae was the most common (46.2%; p=0.035). Similarly, functional outcomes including oral incompetence, ectropion, and trismus were not affected by the extent of resection (p=0.766). However, oral incompetence was higher in patients with tumors originating from oral cavity (p=0.020) and after the performance of mandibulectomy (p=0.003). “
Comment 13: Discussion: 207: Overall survival (OS), Disease free survival (DFS). The same for SCC line 194. Provide the full term once. Check throughout the manuscript
ANSWER and CHANGES 13: We went over the whole manuscript regarding unexplained or unclear abbreviations.
- The abbreviation SCC (squamous cell carcinoma) was already introduced in the introduction (Page 1) and was therefore used consequently throughout the manuscript.
- DFS for disease-free survival has not been explained before and therefore, the corresponding paragraph was changed: Page 7, Lines 226-229: “In fact, they analyzed data of 127 patients with buccal SCCs reporting on significantly better 2-year overall survival (OS: 83.3% vs. 60.1%) and DFS disease-free survival (DFS: 6% vs. 51.9%) in patients undergoing more extensive unit resection compared to conventional surgery.3 “
Comment 14: Line 240: I disagree that multi centric studies are particularly great for studies such as yours. One of the major influencer variables of surgical (especially in surgical oncology) outcomes is the surgeon, hospital, infrastructure. All of the potentially confounding variables cannot be taken into consideration in your study (which is fine per se). But i would not state that this is a strength of your study.
ANSWER 14: Yes you are right. On the one hand a multicentric aspect of outcome studies is desirable to increase patient numbers and the power of your results but on the other hand all of the abovementioned cofounding aspects can decrease the quality of the study.
CHANGES 14: DISCUSSION (Page 9, Lines 269-270): “We believe that the strength of this study lies in its multicentric nature and the analysis of functional outcomes as well as complications.“
Comment 15: Conclusion. highly-required? replace with "necessary". Your conclusion is not great. Please identify some of the findings in your manuscript that you think are really of interest. You can always conclude that there is more to be explored and studied.
ANSWER and CHANGES 15: Conclusion was corrected and adapted as suggested:
- Page 9, 5. Conclusions, Lines 277-283: “More extensive resections with creation of through-and-through defects did not automatically correlate with worse functional outcome or higher recipient site complications. Nonetheless, especially suborbital zone I defects carried an increased risk of wound dehiscence and formation of salivary fistula and requires therefore particular consideration and meticulous reconstruction. However, further prospective studies with homogenous patient cohorts are highly required necessary to define and identify additional factors that may contribute to the functional outcome of patients with carcinomas of the cheek.“
